# A complete hierarchy for the pure state marginal problem in quantum mechanics

Xiao-Dong Yu [1✉], Timo Simnacher [1], Nikolai Wyderka[1,2], H. Chau Nguyen[1] & Otfried Gühne [1]

Clarifying the relation between the whole and its parts is crucial for many problems in science. In quantum mechanics, this question manifests itself in the quantum marginal problem, which asks whether there is a global pure quantum state for some given marginals. This problem arises in many contexts, ranging from quantum chemistry to entanglement theory and quantum error correcting codes. In this paper, we prove a correspondence of the marginal problem to the separability problem. Based on this, we describe a sequence of semidefinite programs which can decide whether some given marginals are compatible with some pure global quantum state. As an application, we prove that the existence of multiparticle absolutely maximally entangled states for a given dimension is equivalent to the separability of an explicitly given two-party quantum state. Finally, we show that the existence of quantum codes with given parameters can also be interpreted as a marginal problem, hence, our complete hierarchy can also be used.

[1] Naturwissenschaftlich-Technische Fakultät, Universität Siegen, Siegen, Germany. [2] Institut für Theoretische Physik III, Heinrich-Heine-Universität Düsseldorf, Düsseldorf, Germany. ✉email: xiao-dong.yu@uni-siegen.de

For a given multiparticle quantum state $|\varphi\rangle$ it is straightforward to compute its marginals or reduced density matrices on some subsets of the particles. The reverse question, whether a given set of marginals is compatible with a global pure state, is, however, not easy to decide. Still, it is at the heart of many problems in quantum physics. Already in the early days, it was a key motivation for Schrödinger to study entanglement[1], and it was recognized as a central problem in quantum chemistry[2]. There, often additional constraints play a role, e.g., if one considers fermionic systems. Then, the anti-symmetry leads to additional constraints on the marginals, generalizing the Pauli principle[3,4]. A variation of the marginal problem is the question of whether or not the marginals determine the global state uniquely or not[5–7]. This is relevant in condensed matter physics, where one may ask whether a state is the unique ground state of a local Hamiltonian[8,9]. Many other cases, such as marginal problems for Gaussian and symmetric states[10,11] and applications in quantum correlation[12], quantum causality[13], and interacting quantum many-body systems[14,15] have been studied.

With the emergence of quantum information processing, various specifications of the marginal problem moved into the center of attention. In entanglement theory, a pure two-particle state is maximally entangled, if the one-particle marginals are maximally mixed. Furthermore, absolutely maximally entangled (AME) states are multiparticle states that are maximally entangled for any bipartition. This makes them valuable ingredients for quantum information protocols[16,17], but it turns out that AME states do not exist for arbitrary dimensions, as not always global states with the desired mixed marginals can be found[18–21]. In fact, also states obeying weaker conditions, where a smaller number of marginals should be maximally mixed, are of fundamental interest, but in general it is open when such states exist[22–24]. More generally, the construction of quantum error-correcting codes, which constitute fundamental building blocks in the design of quantum computer architectures[25–27], essentially amounts to the identification of subspaces of the total Hilbert space, where all states in this space obey certain marginal constraints. This establishes a connection to the AME problem, which consequently was announced to be one of the central problems in quantum information theory[28].

In this paper, we rewrite the marginal problem as an optimization problem over separable states. Here and in the following, the term marginal problem usually refers to the pure state marginal problem in quantum mechanics. This rewriting allows us to transform the nonconvex and thus intractable purity constraint into a complete hierarchy of conditions for a set of marginals to be compatible with a global pure state. Each step is given by a semidefinite program (SDP), the conditions become stronger with each level, and a set of marginals comes from a global state, if and only if all steps are passed. There are at least two advantages of writing the marginal problem as an SDP hierarchy: First, the symmetry in the physical problem can be directly incorporated to drastically simplify the optimization (or feasibility) problem. Second, many known efficient and reliable algorithms are known for solving SDPs[29], which is in stark contrast to nonconvex optimization. To show the effectiveness of our method, we consider the existence problem of AME states. By employing the symmetry, we show that an AME state for a given number of particles and dimension exists, if and only if a specific two-party quantum state is separable. In fact, this allows us to reproduce nearly all previous results on the AME problem[30] with only few lines of calculation. Finally, we show that our approach can also be extended to study the existence problem of quantum codes.

## Results

### Connecting the marginal problem with the separability problem. The formal definition of the marginal problem is the

following: Consider an $n$-particle Hilbert space $\mathcal{H} = \bigotimes_{i=1}^{n} \mathcal{H}_i$, and let $\mathcal{I} \subset \{I | I \subset [n] = \{1, 2, \ldots, n\}\}$ be some subsets of the particles, where the reduced states $\rho_I$ are known marginals. Then, the problem reads

$$
\begin{aligned}
&\text{find} && |\varphi\rangle \\
&\text{s.t.} && \text{Tr}_{I^c}(|\varphi\rangle\langle\varphi|) = \rho_I, \, I \in \mathcal{I}.
\end{aligned}
\tag{1}
$$

Here, $I^c = [n] \setminus I$ denotes the complement of the set $I$. Before explaining our approach, two facts are worth mentioning: First, if the global state $|\varphi\rangle\langle\varphi|$ is not required to be pure, then the quantum marginal problem without purity constraint is already an SDP. Second, if the given marginals are only one-body marginals, that is $\mathcal{I} = \{\{i\} | i \in [n]\}$, the marginals are non-overlapping and the problem in Eq. (1) was solved by Klyachko[31]. For overlapping marginals, however, the solution is more complicated, and this is what we want to discuss in this work.

The main idea of our method is to consider, for a given set of marginals, the compatible states and their extensions to two copies. Then, we can formulate the purity constraint using an SDP. First, let us introduce some notation. Let $\mathcal{C}$ be the set of global states (not necessarily pure) that are compatible with the marginals, i.e.,

$$
\mathcal{C} = \{\rho | \rho \geq 0, \, \text{Tr}_{I^c}(\rho) = \rho_I \quad \forall I \in \mathcal{I}\}.
\tag{2}
$$

Then, we define $\mathcal{C}_2$ to be the convex hull of two copies of the compatible states

$$
\mathcal{C}_2 = \text{conv}\{\rho \otimes \rho | \rho \in \mathcal{C}\} = \left\{\sum_{\mu} p_\mu \rho_\mu \otimes \rho_\mu | \rho_\mu \in \mathcal{C}\right\},
\tag{3}
$$

where the $p_\mu$ form a probability distribution. We denote the two parties as $A$ and $B$, and each of them owns an $n$-body quantum system (see Fig. 1).

To impose the purity constraint, we take advantage of the well-known relation[32]

$$
\text{Tr}(V_{AB}\rho_A \otimes \rho_B) = \text{Tr}(\rho_A \rho_B),
\tag{4}
$$

where $\rho_A$ and $\rho_B$ are arbitrary quantum states, and $V_{AB}$ is the swap operator between parties $A$ and $B$, i.e.,

$$
V_{AB} = \sum_{i,j} |j, i\rangle\langle i, j|,
\tag{5}
$$

which acts on a state $\Phi_{AB} = \sum_{i,j} \omega_{ijkl} |i, j\rangle\langle k, l|$ as $V_{AB}\Phi_{AB} = \sum_{i,j} \omega_{ijkl} |j, i\rangle\langle k, l|$. For a state $\Phi_{AB}$ in $\mathcal{C}_2$ this implies that

$$
\text{Tr}(V_{AB}\Phi_{AB}) = \sum_{\mu} p_\mu \text{Tr}(\rho_\mu^2) \leq 1.
\tag{6}
$$

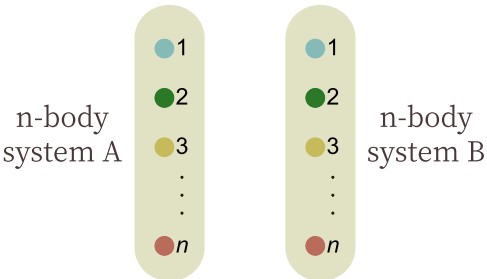

**Fig. 1 Two-party extension for the marginal problem.** In the marginal problem, one aims to characterize the pure states $|\varphi\rangle$ on $n$ particles, which are compatible with given marginals. The key idea of our approach is to drop the purity constraint and to consider mixed states $\rho$ with the given marginals. Then, the purity is enforced by considering a two-party extension $\Phi_{AB}$.

Furthermore, equality in Eq. (6) is attained if and only if all $\rho_\mu$ are pure states. This leads to our first key observation: there exists a pure state in $\mathcal{C}$ if and only if $\max_{\Phi_{AB} \in \mathcal{C}_2} \mathrm{Tr}(V_{AB}\Phi_{AB}) = 1$.

What remains to be done is the characterization of the set $\mathcal{C}_2$, then we can formulate the quantum marginal problem as an optimization problem over this set. This can be done by taking advantage of the separability property[32] of the states in $\mathcal{C}_2$ with respect to the bipartition $(A|B)$.

**Theorem 1** *There exists a pure quantum state $|\varphi\rangle$ that satisfies $\mathrm{Tr}_{I^c}(|\varphi\rangle\langle\varphi|) = \rho_I$ for all $I \in \mathcal{I}$ if, and only if, the solution of the following convex optimization is equal to one,*

$$\max_{\Phi_{AB}} \ \mathrm{Tr}(V_{AB}\Phi_{AB}) \tag{7}$$

$$\text{s.t. } \Phi_{AB} \in \mathrm{SEP}, \ \mathrm{Tr}(\Phi_{AB}) = 1, \tag{8}$$

$$\mathrm{Tr}_{A_{I^c}B_{I^c}}(\Phi_{AB}) = \rho_I \otimes \rho_I \quad \forall I \in \mathcal{I}, \tag{9}$$

*where SEP denotes the set of separable states w.r.t. the bipartition $(A|B)$, $A_{I^c}$ denotes all subsystems $A_i$ for $i \in I^c$, and similarly for $B_{I^c}$.*

**Proof** On the one hand, if there exists a pure state $|\varphi\rangle\langle\varphi| \in \mathcal{C}$, one can easily verify that $\Phi_{AB} = |\varphi\rangle\langle\varphi| \otimes |\varphi\rangle\langle\varphi|$ satisfies the constraints in Eqs. (8) and (9) as well as $\mathrm{Tr}(V_{AB}\Phi_{AB}) = 1$.

On the other hand, if the solution of Eq. (7) is equal to one, then the separability constraint and Eq. (6) imply that $\Phi_{AB}$ must be of the form[33]

$$\Phi_{AB} = \sum_\mu p_\mu |\psi_\mu\rangle\langle\psi_\mu| \otimes |\psi_\mu\rangle\langle\psi_\mu|. \tag{10}$$

Writing $\mathrm{Tr}_{I^c}(|\psi_\mu\rangle\langle\psi_\mu|) = \sigma_I^{(\mu)}$, the constraint in Eq. (9) implies that

$$\sum_\mu p_\mu \sigma_I^{(\mu)} \otimes \sigma_I^{(\mu)} = \rho_I \otimes \rho_I \quad \forall I \in \mathcal{I}. \tag{11}$$

Without loss of generality, we assume that all $p_\mu$ are strictly positive and we want to show that all $\sigma_I^{(\mu)} = \rho_I$, which will imply that each $|\psi_\mu\rangle$ is a pure state with the desired marginals. Let $X$ be any Hermitian matrix such that $\mathrm{Tr}(X\rho_I) = 0$, then Eq. (11) and the relation $\mathrm{Tr}[(X \otimes X)(\sigma \otimes \sigma)] = [\mathrm{Tr}(X\sigma)]^2$ imply that

$$\sum_\mu p_\mu \left[\mathrm{Tr}\left(X\sigma_I^{(\mu)}\right)\right]^2 = [\mathrm{Tr}(X\rho_I)]^2 = 0. \tag{12}$$

By noting that $\mathrm{Tr}(X\sigma_I^{(\mu)})$ are real numbers, we get that $\mathrm{Tr}(X\sigma_I^{(\mu)}) = 0$, for all $\mu$ and all $X$ such that $\mathrm{Tr}(X\rho_I) = 0$. Thus, all $\sigma_I^{(\mu)}$ are proportional to $\rho_I$ and, together with $\mathrm{Tr}(\rho_I) = \mathrm{Tr}(\sigma_I^{(\mu)}) = 1$, this implies that $\sigma_I^{(\mu)} = \rho_I$ for all $\mu$.

Before proceeding further, we would like to add a few remarks. First, in Theorem 1 the constraint in Eq. (9) can be replaced by a stronger condition

$$\mathrm{Tr}_{A_{I^c}}(\Phi_{AB}) = \rho_I \otimes \mathrm{Tr}_A(\Phi_{AB}) \quad \forall I \in \mathcal{I}. \tag{13}$$

This is because for any (not necessarily separable) quantum state $\Phi_{AB}$ satisfying $\mathrm{Tr}(V_{AB}\Phi_{AB}) = 1$, Eq. (13) implies the validity of Eq. (9). Hence, this replacement will lead to an equivalent result as in Theorem 1. However, when considering relaxations of the optimization in Eq. (7) by replacing the separability constraint in Eq. (8) with some entanglement criteria, Eq. (13) may be strictly stronger for certain marginal problems.

Second, if one finds that $\mathrm{Tr}(V_{AB}\Phi_{AB}) = 1$, this is equivalent to $V_{AB}\Phi_{AB} = \Phi_{AB}$, as the largest eigenvalue of $V_{AB}$ is one. Physically, this means that $\Phi_{AB}$ is a two-party state acting on the symmetric subspace only. Hence, Theorem 1 is also equivalent to the

feasibility problem

$$\text{find} \quad \Phi_{AB} \in \mathrm{SEP} \tag{14}$$

$$\text{s.t.} \quad V_{AB}\Phi_{AB} = \Phi_{AB}, \ \mathrm{Tr}(\Phi_{AB}) = 1, \tag{15}$$

$$\mathrm{Tr}_{A_{I^c}}(\Phi_{AB}) = \rho_I \otimes \mathrm{Tr}_A(\Phi_{AB}) \quad \forall I \in \mathcal{I}. \tag{16}$$

Furthermore, any feasible state $\Phi_{AB}$ can be used for constructing the global state $|\varphi\rangle$ with the desired marginals, as the proof of Theorem 1 implies any pure state in the separable decomposition of $\Phi_{AB}$ can give a desired global state.

Third, the separability condition in the optimization Eq. (8) is usually not easy to characterize, hence relaxations of the problem need to be considered. The first candidate is the positive partial transpose (PPT) criterion[34,35], which is an SDP relaxation of the optimization in Eq. (7). The PPT relaxation provides a pretty good approximation when the local dimension and the number of parties are small. In the following, inspired by the symmetric extension criterion[36], we propose a multi-party extension method and obtain a complete hierarchy for the marginal problem.

**The hierarchy for the marginal problem.** In order to generalize Theorem 1, we first need to extend $\mathcal{C}_2$ in Eq. (3) from two to an arbitrary number of copies of $\rho$. That is, we define $\mathcal{C}_N = \mathrm{conv}\{\rho^{\otimes N} | \rho \in \mathcal{C}\}$. Second, we introduce the notion of the symmetric subspace. We denote the $N$ parties as $A, B, \ldots, Z$, and each of them owns an $n$-body quantum system. For any $\mathcal{H}^{\otimes N} := \mathcal{H}_A \otimes \mathcal{H}_B \otimes \cdots \otimes \mathcal{H}_Z$, the symmetric subspace is defined as

$$\left\{|\Psi\rangle \in \mathcal{H}^{\otimes N} \mid V_\Sigma|\Psi\rangle = |\Psi\rangle \quad \forall \Sigma \in S_N\right\}, \tag{17}$$

where $S_N$ is the permutation group over $N$ symbols and $V_\Sigma$ are the corresponding operators on the $N$ parties $A, B, \ldots, Z$ (see Fig. 2). Let $P_N^+$ denote the orthogonal projector onto the symmetric subspace of $\mathcal{H}^{\otimes N}$. $P_N^+$ can be explicitly written as

$$P_N^+ = \frac{1}{N!} \sum_{\Sigma \in S_N} V_\Sigma. \tag{18}$$

In particular, for two parties we have the well-known relation $P_2^+ = (\mathbb{1}_{AB} + V_{AB})/2$, which implies that $\mathrm{Tr}(V_{AB}\Phi_{AB}) = 1$ if and only if $\mathrm{Tr}(P_2^+\Phi_{AB}) = 1$. Also, $V_{AB}\Phi_{AB} = \Phi_{AB}$ is equivalent to $P_2^+\Phi_{AB}P_2^+ = \Phi_{AB}$. Hereafter, without ambiguity, we will use $P_N^+$ to

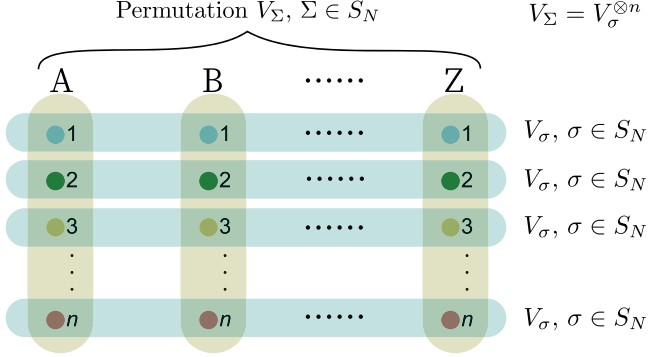

**Fig. 2 Complete hierarchy for the marginal problem.** In order to formulate the hierarchy for the marginal problem, one extends the two copies in Fig. 1 to an arbitrary number of copies $N$. If the marginal problem has a solution $|\varphi\rangle$, then there are multi-party extensions $\Phi_{AB\cdots Z}$ in the symmetric subspace specified by $V_\Sigma = V_\sigma^{\otimes n}$ for any number of copies, obeying some semidefinite constraints.

denote both the symmetric subspace and the corresponding orthogonal projector.

Suppose that there exists a pure state $\rho \in \mathcal{C}$. It is easy to see that $\Phi_{AB\cdots Z} = \rho^{\otimes N}$ satisfies

$$P_N^+ \Phi_{AB\cdots Z} P_N^+ = \Phi_{AB\cdots Z}, \tag{19}$$

$$\Phi_{AB\cdots Z} \in \text{SEP}, \ \text{Tr}(\Phi_{AB\cdots Z}) = 1, \tag{20}$$

$$\text{Tr}_{A_{I^c}}(\Phi_{AB\cdots Z}) = \rho_I \otimes \text{Tr}_A(\Phi_{AB\cdots Z}) \quad \forall I \in \mathcal{I}. \tag{21}$$

Here, the separability can be understood as either full separability or biseparability, since they are equivalent in the symmetric subspace[33]. Relaxing $\Phi_{AB\cdots Z} \in \text{SEP}$, we obtain a complete hierarchy for the quantum marginal problem:

**Theorem 2** *There exists a pure quantum state $|\varphi\rangle$ that satisfies $\text{Tr}_{I^c}(|\varphi\rangle\langle\varphi|) = \rho_I$ for all $I \in \mathcal{I}$ if and only if for all $N \geq 2$ there exists an N-party quantum state $\Phi_{AB\cdots Z}$ such that*

$$P_N^+ \Phi_{AB\cdots Z} P_N^+ = \Phi_{AB\cdots Z}, \tag{22}$$

$$\Phi_{AB\cdots Z} \geq 0, \ \text{Tr}(\Phi_{AB\cdots Z}) = 1, \tag{23}$$

$$\text{Tr}_{A_{I^c}}(\Phi_{AB\cdots Z}) = \rho_I \otimes \text{Tr}_A(\Phi_{AB\cdots Z}) \quad \forall I \in \mathcal{I}. \tag{24}$$

*Each step of this hierarchy is a semidefinite feasibility problem, and the conditions become more restrictive if N increases.*

The proof of Theorem 2 is shown in the "Methods" section. Notably, we can add any criterion of full separability, e.g., the PPT criterion for all bipartitions, as extra constraints to the feasibility problem. Then, Theorem 2 still provides a complete hierarchy for the quantum marginal problem. In addition, the quantum marginal problems of practical interest are usually highly symmetric. These symmetries can be utilized to largely simplify the problems in Theorems 1 and 2. Indeed, taking advantage of symmetries is usually necessary for practical applications, because the general quantum marginal problem is QMA-complete[37,38]. Notably, even for non-overlapping marginals, despite recent progress in refs. [39–41], it is still an open problem whether there exists a polynomial-time algorithm. In the following, we illustrate how symmetry can drastically simplify quantum marginal problems with the existence problem of AME states.

**Absolutely maximally entangled states**. We first recall the definition of AME states. An $n$-qudit state $|\psi\rangle$ is called an AME state, denoted as $\text{AME}(n, d)$, if it satisfies

$$\text{Tr}_{I^c}(|\psi\rangle\langle\psi|) = \frac{\mathbb{1}_{d^r}}{d^r} \quad \forall I \in \mathcal{I}_r, \tag{25}$$

where $\mathcal{I}_r = \{I \subset [n] \mid |I| = r\}$ and $r = \lfloor n/2 \rfloor$. Thus, Eq. (14) implies that an $\text{AME}(n, d)$ exists if and only if the following problem is feasible,

$$\text{find } \Phi_{AB} \in \text{SEP} \tag{26}$$

$$\text{s.t. } \text{Tr}(\Phi_{AB}) = 1, \ V_{AB}\Phi_{AB} = \Phi_{AB}, \tag{27}$$

$$\text{Tr}_{A_{I^c}}(\Phi_{AB}) = \frac{\mathbb{1}_{d^r}}{d^r} \otimes \text{Tr}_A(\Phi_{AB}) \quad \forall I \in \mathcal{I}_r. \tag{28}$$

Direct evaluation of the problem is usually difficult, because the dimension of $\Phi_{AB}$ is $d^{2n} \times d^{2n}$, which is already very large for the simplest cases. For instance, for the 4-qubit case, the size of $\Phi_{AB}$ is $256 \times 256$.

To resolve this size issue, we investigate the symmetries that can be used to simplify the feasibility problem. Let $\mathcal{X}$ denotes the set of $\Phi_{AB}$ that satisfies the constraints in Eqs. (26), (27), and (28). If we find a unitary group $G$ such that for all $g \in G$ and $\Phi_{AB} \in \mathcal{X}$

we have that

$$g\Phi_{AB}g^\dagger \in \mathcal{X}. \tag{29}$$

Then, the convexity of $\mathcal{X}$ implies that we can add a symmetry constraint to the constraints in Eqs. (26), (27), and (28), namely,

$$g\Phi_{AB}g^\dagger = \Phi_{AB} \quad \forall g \in G. \tag{30}$$

In the following, we will show that the symmetries of the set of AME states (if they exist for given $n$ and $d$) are restrictive enough to leave only a single unique candidate for $\Phi_{AB}$, for which separability needs to be checked. The set of $\text{AME}(n, d)$ is invariant under local unitaries and permutations on the $n$ particles, so by Theorem 1 (or by direct verification) the following two classes of unitaries satisfy Eq. (29),

$$U_1 \otimes \cdots \otimes U_n \otimes U_1 \otimes \cdots \otimes U_n \quad \forall U_i \in SU(d), \tag{31}$$

$$\pi \otimes \pi \quad \forall \pi \in S_n, \tag{32}$$

where $\pi = \pi(A_1, A_2, ..., A_n) = \pi(B_1, B_2, ..., B_n)$ denotes the permutation operators on $\mathcal{H}_A$ and $\mathcal{H}_B$. Note that the $U_i$ in Eq. (31) can be different.

First, let us view $V_{AB}$ and $\Phi_{AB}$ as $V_{12...n}$ and $\Phi_{12...n}$, where $i$ labels the subsystems $A_iB_i$. Hereafter, without ambiguity, we will omit the subscripts of

$$\mathbb{1} := \mathbb{1}_{d^2}, \quad V := V_{A_iB_i}, \tag{33}$$

for simplicity. From this perspective, $V_{AB}$ can be written as $V^{\otimes n}$, and the symmetries in Eqs. (31) and (32) can be written as $\bigotimes_{i=1}^n (U_i \otimes U_i)$ for $U_i \in SU(d)$ and $\Pi = \Pi(A_1B_1, A_2B_2, ..., A_nB_n)$ for $\Pi \in S_n$, respectively. According to Werner's result[32], a $(U \otimes U)$-invariant Hermitian operator must be of the form $\alpha \mathbb{1} + \beta V$ with $\alpha, \beta \in \mathbb{R}$. This implies that a $\left[\bigotimes_{i=1}^n (U_i \otimes U_i)\right]$-invariant state must be a linear combination of operators of the form

$$\bigotimes_{i=1}^n (\alpha_i \mathbb{1} + \beta_i V) \quad \forall \alpha_i, \beta_i \in \mathbb{R}. \tag{34}$$

In addition, we take advantage of the permutation symmetry under $\Pi \in S_n$ to write any invariant $\Phi_{AB}$ as

$$\Phi_{AB} = \sum_{i=0}^n x_i \mathcal{P}\{V^{\otimes i} \otimes \mathbb{1}^{\otimes (n-i)}\}, \tag{35}$$

where $\mathcal{P}$ represents the sum over all possible permutations that give different terms, e.g., $\mathcal{P}\{V \otimes \mathbb{1} \otimes \mathbb{1}\} = V \otimes \mathbb{1} \otimes \mathbb{1} + \mathbb{1} \otimes V \otimes \mathbb{1} + \mathbb{1} \otimes \mathbb{1} \otimes V$.

Inserting this ansatz in Eqs. (27) and (28) one can show by brute force calculation that the $x_i$ are uniquely determined and given by

$$x_i = \frac{(-1)^i}{(d^2-1)^n} \sum_{l=0}^n \sum_{k=0}^l \frac{(-1)^l \binom{i}{k}\binom{n-i}{l-k}}{\min\{d^{i+2l-2k}, d^{n+i-2k}\}}, \tag{36}$$

where we use the convention that $\binom{i}{j} = 0$ when $j < 0$ or $j > i$; see Supplementary Note 1 for details. This means that the two-party extension under the symmetries is independent of the specific AME state, which is an interesting structural result considering that there exist even infinite families of $\text{AME}(n, d)$ states that are not SLOCC equivalent[42]. Together with Theorem 1, this result implies that an AME state exists if and only if $\Phi_{AB}$ is a separable quantum state.

**Theorem 3** *An $\text{AME}(n, d)$ state exists if and only if the operator $\Phi_{AB}$ defined by Eqs. (35) and (36) is a separable state w.r.t. the bipartition $(A|B) = (A_1A_2...A_n|B_1B_2...B_n)$.*

To check the separability of $\Phi_{AB}$, we first consider the positivity condition and the PPT condition. It is easy to see that $\Phi_{AB}$ can be

written as

$$\Phi_{AB} = \sum_{i=0}^{n} p_i \mathcal{P}\left\{P_+^{\otimes(n-i)} \otimes P_-^{\otimes i}\right\}, \qquad (37)$$

and $\Phi_{AB}^{T_B}$ can be written as

$$\Phi_{AB}^{T_B} = \sum_{i=0}^{n} q_i \mathcal{P}\left\{P_\phi^{\otimes(n-i)} \otimes P_\perp^{\otimes i}\right\}, \qquad (38)$$

where

$$P_\pm = \frac{1}{2}(\mathbb{1} \pm V), \ P_\phi = |\phi^+\rangle\langle\phi^+|, \ P_\perp = \mathbb{1} - P_\phi, \qquad (39)$$

with $|\phi^+\rangle = \frac{1}{\sqrt{d}}\sum_{k=1}^{d}|k\rangle|k\rangle$. Here $p_i$ and $q_i$ are the eigenvalues of $\Phi_{AB}$ and $\Phi_{AB}^{T_B}$, respectively. Then, we can simplify the positivity condition $\Phi_{AB} \geq 0$ and the PPT condition $\Phi_{AB}^{T_B} \geq 0$ to

$$\sum_{l=0}^{n}\sum_{k=0}^{l} \frac{(-1)^k \binom{i}{k}\binom{n-i}{l-k}}{\min\{d^l, d^{n-l}\}} \geq 0, \qquad (40)$$

$$\sum_{k=0}^{i} \frac{(-1)^k \binom{i}{k}}{\min\{d^{2(n+k-i)}, d^n\}} \geq 0, \qquad (41)$$

for all $i = 0, 1, 2, \ldots, n$. Note that the latter inequality is trivial for $i \leq r$.

The explicit form of $p_i$ and $q_i$ and the proof of the conditions in Eqs. (40) and (41) are shown in Supplementary Note 2. The positivity and PPT conditions can already rule out the existence of many AME states. Actually, they can reproduce all the known nonexistence results[30] except AME(7, 2)[20]. To get a higher-order approximation, we provide a general framework for performing the symmetric extension in Supplementary Notes 3 and 4.

As the open problem of the existence of AME(4, 6) is of particular interest in the quantum information community[28,43], we explicitly express it as the following corollary.

**Corollary 1** *An* AME(4, 6) *state exists if and only if the quantum state*

$$\Phi_{AB} = \frac{1}{2 \cdot 6^4}\left(\frac{P_+^{\otimes 4}}{343} + \frac{\mathcal{P}\{P_+^{\otimes 2} \otimes P_-^{\otimes 2}\}}{315} + \frac{P_-^{\otimes 4}}{375}\right), \qquad (42)$$

*is separable, or equivalently,*

$$\Phi_{AB}^{T_B} = \frac{1}{6^4}\left(P_\phi^{\otimes 4} + \frac{\mathcal{P}\{P_\phi \otimes P_\perp^{\otimes 3}\}}{35^2} + \frac{33 P_\perp^{\otimes 4}}{35^3}\right), \qquad (43)$$

*is separable w.r.t. bipartition* $(A|B)$.

At the moment, we are unable to decide separability of these states; in Supplementary Note 5 we provide a short discussion of this problem.

**Quantum codes**. As another application, we show that our method can also be used to analyze the existence of quantum error-correcting codes. For simplicity, we only consider pure quantum codes[44] in the text; see "Methods" for the general case. Our starting point is the fact that pure quantum codes are closely related to $m$-uniform states[18]. More precisely, an $((n, K, m+1))_d$ pure code exists if and only if there exists a $K$-dimensional subspace $\mathcal{Q}$ of $\mathcal{H} = \bigotimes_{i=1}^{n} \mathcal{H}_i = (\mathbb{C}^d)^{\otimes n}$ such that all states in $\mathcal{Q}$ are $m$-uniform, i.e., for all $|\varphi\rangle \in \mathcal{Q}$

$$\mathrm{Tr}_{I^c}(|\varphi\rangle\langle\varphi|) = \frac{\mathbb{1}_{d^m}}{d^m} \quad \forall I \in \mathcal{I}_m, \qquad (44)$$

where $\mathcal{I}_m = \{I \in [n] \,|\, |I| = m\}$ and $I^c = [n] \backslash I$. The existence of $((n, 1, m+1))_d$ pure codes reduces to the existence of $m$-uniform

states, for which the methods from the last section are directly applicable. Here, we show that the existence of $((n, K, m+1))_d$ pure codes can still be written as a marginal problem if $K > 1$. To do so, we define an auxiliary system $\mathcal{H}_0 = \mathbb{C}^K$ and let $\widetilde{\mathcal{H}} = \mathcal{H}_0 \otimes \mathcal{H} = \bigotimes_{i=0}^{n} \mathcal{H}_i = \mathbb{C}^K \otimes (\mathbb{C}^d)^{\otimes n}$. Now, we can write the existence of $((n, K, m+1))_d$ pure codes as a marginal problem on $\widetilde{\mathcal{H}}$.

**Lemma 1** *A quantum* $((n, K, m+1))_d$ *pure code exists if and only if there exists a quantum state* $|Q\rangle$ *in* $\widetilde{\mathcal{H}}$ *such that*

$$\mathrm{Tr}_{I^c}(|Q\rangle\langle Q|) = \frac{\mathbb{1}_{Kd^m}}{Kd^m} \quad \forall I \in \mathcal{I}_m, \qquad (45)$$

*where* $I^c$ *is still defined as* $\{1, 2, \ldots, n\} \backslash I$.

**Proof** We first show the necessity part. Suppose that a $((n, K, m+1))_d$ code with corresponding subspace $\mathcal{Q}$ exists. We define an entangled state $|Q\rangle$ in $\mathcal{H}_0 \otimes \mathcal{Q} \subset \widetilde{\mathcal{H}}$ as

$$|Q\rangle = \frac{1}{\sqrt{K}}\sum_{k=1}^{K}|k\rangle|k_L\rangle, \qquad (46)$$

where $\{|k\rangle\}_{k=1}^{K}$ and $\{|k_L\rangle\}_{k=1}^{K}$ are orthonormal bases for $\mathcal{H}_0$ and $\mathcal{Q}$, respectively. Then for any pure state $|a\rangle$ in $\mathcal{H}_0$, $\sqrt{K}\langle a|Q\rangle \in \mathcal{Q}$. Hence, Eq. (44) implies that

$$\mathrm{Tr}_0[\mathrm{Tr}_{I^c}(|a\rangle\langle a| \otimes \mathbb{1}_{d^n}|Q\rangle\langle Q|)] = \frac{\mathbb{1}_{d^m}}{Kd^m} \quad \forall I \in \mathcal{I}_m, \qquad (47)$$

for all $|a\rangle$ in $\mathcal{H}_0$, which in turn implies Eq. (45).

To prove the sufficiency part, let $\mathcal{Q}$ be the space generated by the pure states $|\varphi_a\rangle = \sqrt{K}\langle a|Q\rangle$ for all $|a\rangle$ in $\mathcal{H}_0$. Then, Eq. (45) implies that all $|\varphi_a\rangle$ are $m$-uniform states. Furthermore, from $\mathrm{rank}(\mathrm{Tr}_0(|Q\rangle\langle Q|)) = \mathrm{rank}(\mathrm{Tr}_{12\cdots n}(|Q\rangle\langle Q|)) = \mathrm{rank}(\mathbb{1}_K/K) = K$ it follows that $\mathcal{Q}$ is a $K$-dimensional subspace.

Thus, Theorem 1 gives a necessary and sufficient condition for the existence of $((n, K, m+1))_d$ pure codes.

**Proposition 1** *A quantum* $((n, K, m+1))_d$ *pure code exists if and only if there exists* $\Phi_{AB}$ *in* $\widetilde{\mathcal{H}}_A \otimes \widetilde{\mathcal{H}}_B = [\mathbb{C}^K \otimes (\mathbb{C}^d)^{\otimes n}]^{\otimes 2}$ *such that*

$$\Phi_{AB} \in \mathrm{SEP}, \ V_{AB}\Phi_{AB} = \Phi_{AB}, \ \mathrm{Tr}(\Phi_{AB}) = 1, \qquad (48)$$

$$\mathrm{Tr}_{A_{I^c}}(\Phi_{AB}) = \frac{\mathbb{1}_{Kd^m}}{Kd^m} \otimes \mathrm{Tr}_A(\Phi_{AB}) \quad \forall I \in \mathcal{I}_m, \qquad (49)$$

*where SEP denotes the set of separable states w.r.t. the bipartition* $(A|B) = (A_0 A_1 \cdots A_n | B_0 B_1 \cdots B_n)$, $V_{AB}$ *is the swap operator between* $\widetilde{\mathcal{H}}_A$ *and* $\widetilde{\mathcal{H}}_B$, *and* $A_{I^c}$ *denotes all subsystems* $A_i$ *for* $i \in I^c$.

Furthermore, the multi-party extension and symmetrization techniques that we developed for AME states can be easily adapted to the quantum error-correcting codes. For instance, the PPT relaxation can be written as a linear program and the symmetric extensions can be written as SDPs. An important difference is that the symmetrized $\Phi_{AB}$ for quantum error-correcting codes is no longer uniquely determined by the marginals in general. Finally, we would like to mention that Lemma 5 is of independent interest on its own. For example, Eq. (45) implies that $Kd^m \leq \sqrt{Kd^n}$, as $\mathrm{rank}(\mathrm{Tr}_{I^c}(|Q\rangle\langle Q|)) \leq \sqrt{\dim(\widetilde{\mathcal{H}})}$. This provides a simple proof for the quantum Singleton bound[44,45] $K \leq d^{n-2m}$ for pure codes.

## Discussion

We have shown that the marginal problem for multiparticle quantum systems is closely related to the problem of entanglement and separability for two-party systems. More precisely, we have shown that the existence of a pure multiparticle state with given marginals can be reformulated as the existence of a two-

party separable state with additional semidefinite constraints. This allows for further refinements: First, one may use the multi-party extension technique to develop a complete hierarchy for the quantum marginal problem. Second, one can use symmetries of the original marginal problem, to restrict the search of the two-party separable state further. For the AME problem, this allows us to determine a unique candidate for the state, and it remains to check its separability properties. Finally, the approach can be extended to characterize the existence of quantum codes.

Our work provides new insights into several subfields of quantum information theory. First, it may provide a significant step towards solving the problem of the existence of the AME(4, 6) state or quantum orthogonal Latin squares, a problem which has been highlighted as an outstanding problem in quantum information theory[28]. Second, there are already a variety of results on the separability problem, and in the future, these can be used to study marginal problems in various situations. Finally, it would be interesting to extend our work to other versions of the marginal problem, e.g., in fermionic systems or with a relaxed version of the purity constraint. We believe that our approach can also lead to progress in these cases.

## Methods

**Proof of Theorem 2.** To prove Theorem 2, we take advantage of the following lemma, which can be viewed as a special case of the quantum de Finetti theorem[46].

**Lemma 2** *Let $\rho_N$ be an $N$-party quantum state in the symmetric subspace $P_N^+$, then there exists a $k$-party quantum state*

$$\sigma_k = \sum_\mu p_\mu |\psi_\mu\rangle\langle\psi_\mu|^{\otimes k}, \tag{50}$$

*i.e., a fully separable state in $P_k^+$, such that*

$$\left\|\mathrm{Tr}_{N-k}(\rho_N) - \sigma_k\right\| \le \frac{4kD}{N}, \tag{51}$$

*where $\|\cdot\|$ is the trace norm and $D$ is the local dimension.*

The necessity part of Theorem 2 is obvious. Hence, we only need to prove the sufficiency part, i.e., that the existence of an $N$-party quantum state $\Phi_{AB\cdots Z}^N$ for arbitrary $N$ implies the existence of $|\varphi\rangle$. Let $\Phi_{AB}^N = \mathrm{Tr}_{C\cdots Z}(\Phi_{ABC\cdots Z})$, then $\Phi_{AB}^N$ satisfies

$$\mathrm{Tr}(\Phi_{AB}^N) = 1, \quad \mathrm{Tr}_{A_{I^c}}(\Phi_{AB}^N) = \rho_I \otimes \mathrm{Tr}_A(\Phi_{AB}^N) \quad \forall I \in \mathcal{I}. \tag{52}$$

Further, Lemma 7 implies that there exist separable states $\widetilde{\Phi}_{AB}^N$ such that

$$V_{AB}\widetilde{\Phi}_{AB}^N = \widetilde{\Phi}_{AB}^N, \tag{53}$$

$$\left\|\Phi_{AB}^N - \widetilde{\Phi}_{AB}^N\right\| \le \frac{8D}{N}. \tag{54}$$

As the set of quantum states for any fixed dimension is compact, we can choose a convergent subsequence $\Phi_{AB}^{N_i}$ of the sequence $\Phi_{AB}^N$. Thus, Eq. (54) implies that

$$\Phi_{AB} := \lim_{i\to+\infty} \Phi_{AB}^{N_i} = \lim_{i\to+\infty} \widetilde{\Phi}_{AB}^{N_i}. \tag{55}$$

Thus, Eqs. (52) and (53) and the fact that the set of separable states is closed imply that $\Phi_{AB}$ satisfies all constraints in Eqs. (14), (15), and (16). Then, Theorem 2 follows directly from Theorem 1.

**General quantum codes.** In general, a quantum $((n, K, m+1))_d$ code exists if and only if there exists a $K$-dimensional subspace $\mathcal{Q}$ of $\mathcal{H} = \bigotimes_{i=1}^n \mathcal{H}_i = (\mathbb{C}^d)^{\otimes n}$ such that for all $|\varphi\rangle \in \mathcal{Q}$

$$\mathrm{Tr}_{I^c}(|\varphi\rangle\langle\varphi|) = \rho_I \quad \forall I \in \mathcal{I}_m, \tag{56}$$

where $\rho_I$ are marginals that are arbitrary but independent of $|\varphi\rangle$, $\mathcal{I}_m = \{I \in [n] | |I| = m\}$, and $I^c = [n]\backslash I = \{1, 2, \ldots, n\}\backslash I$. Similar to the case of pure codes, we can prove the following lemma.

**Lemma 3** *A quantum $((n, K, m+1))_d$ code exists if and only if there exists a quantum state $|Q\rangle$ in $\widetilde{\mathcal{H}}$ and marginal states $\rho_I$ such that*

$$\mathrm{Tr}_{I^c}(|Q\rangle\langle Q|) = \frac{\mathbb{1}_K}{K} \otimes \rho_I \quad \forall I \in \mathcal{I}_m, \tag{57}$$

where $\widetilde{\mathcal{H}} = \mathcal{H}_0 \otimes \mathcal{H} = \bigotimes_{i=0}^n \mathcal{H}_i = \mathbb{C}^K \otimes (\mathbb{C}^d)^{\otimes n}$ and $I^c$ is defined as $[n]\backslash I = \{1, 2, \ldots, n\}\backslash I$.

If the marginals $\rho_I$ are given like in the case of pure codes, the problem reduces to a marginal problem. However, to ensure the existence of $((n, K, m+1))_d$ codes, an arbitrary set of marginals is sufficient. This makes the problem no longer a marginal problem, however, we can circumvent this issue by observing that Eq. (57) is equivalent to

$$\mathrm{Tr}_0[(M_0 \otimes \mathbb{1}_I)\mathrm{Tr}_{I^c}(|Q\rangle\langle Q|)] = 0 \quad \forall I \in \mathcal{I}_m, \tag{58}$$

for all $M_0$ such that $\mathrm{Tr}(M_0) = 0$. Moreover, we can choose an arbitrary basis $\mathcal{B}$ for $\{M_0 | \mathrm{Tr}(M_0) = 0, M_0^\dagger = M_0\}$. Then, with the general result on rank-constrained optimization from ref. [47], we obtain the following theorem, and similar to the AME existence problem, a complete hierarchy can be constructed using the symmetric extension technique.

**Proposition 2** *A quantum $((n, K, m+1))_d$ code exists if and only if there exists $\Phi_{AB}$ in $\widetilde{\mathcal{H}}_A \otimes \widetilde{\mathcal{H}}_B = [\mathbb{C}^K \otimes (\mathbb{C}^d)^{\otimes n}]^{\otimes 2}$ such that*

$$\Phi_{AB} \in \mathrm{SEP}, \quad V_{AB}\Phi_{AB} = \Phi_{AB}, \quad \mathrm{Tr}(\Phi_{AB}) = 1, \tag{59}$$

$$\mathrm{Tr}_{A_0} \mathrm{Tr}_{A_{I^c}} [(M_{A_0} \otimes \mathbb{1}_{A_0^c})\Phi_{AB}] = 0, \tag{60}$$

*for all $I \in \mathcal{I}_m$ and $M_{A_0} \in \mathcal{B}$, where the SEP means the separability with respect to the bipartition $(A|B) = (A_0 A_1 \cdots A_n | B_0 B_1 \cdots B_n)$, $V_{AB}$ is the swap operator between $\widetilde{\mathcal{H}}_A$ and $\widetilde{\mathcal{H}}_B$, $A_{I^c}$ denotes all subsystems $A_i$ for $i \in I^c$, and $\mathbb{1}_{A_0^c}$ denote the identity operator on $AB\backslash A_0 = A_1 A_2 \cdots A_n B_0 B_1 B_2 \cdots B_n$.*

By noticing that the set of $((n, K, m+1))_d$ (pure or general) codes, or rather, the set of states $|Q\rangle$, is invariant under local unitaries and permutations on the bodies $123\cdots n$, we can assume that $\Phi_{AB}$ is invariant under the following two classes of unitaries

$$U_0 \otimes U_1 \otimes \cdots \otimes U_n \otimes U_0 \otimes U_1 \otimes \cdots \otimes U_n, \tag{61}$$

$$\mathrm{id}_0 \otimes \pi \otimes \mathrm{id}_0 \otimes \pi. \tag{62}$$

for all $U_0 \in SU(K)$, $U_i \in SU(d)$, and $\pi \in S_n$. Thus, the symmetrized $\Phi_{AB}$ is of the form

$$\begin{aligned}\Phi_{AB} = \mathbb{1}_{K^2} \otimes \sum_{i=0}^n x_i \mathcal{P}\{V^{\otimes i} \otimes \mathbb{1}^{\otimes(n-i)}\} \\ + V_{A_0 B_0} \otimes \sum_{i=0}^n y_i \mathcal{P}\{V^{\otimes i} \otimes \mathbb{1}^{\otimes(n-i)}\},\end{aligned} \tag{63}$$

for $x_i, y_i \in \mathbb{R}$. Hence, all the techniques we developed for AME states can be easily adapted to the quantum error-correcting codes. For example, the PPT relaxation can be written as a linear program and the symmetric extension can be written as SDPs.

## Data availability

Data sharing not applicable to this article as no datasets were generated or analyzed during the current study.

## Code availability

The codes used for this study are available from the corresponding author upon reasonable request.

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

## Acknowledgements

We would like to thank Felix Huber and Géza Tóth for discussions. This work was supported by the Deutsche Forschungsgemeinschaft (DFG, German Research Foundation - 447948357), the ERC (Consolidator Grant 683107/TempoQ), and the House of Young Talents Siegen. N.W. acknowledges support by the QuantERA grant QuICHE and the German ministry of education and research (BMBF grant no. 16KIS1119K).

## Author contributions

X.-D.Y., T.S., N.W., H.C.N., and O.G. participated in deriving the results and writing the manuscript. O.G. supervised the project.

## Funding

## Competing interests

The authors declare no competing interests.
