## [Peer Review File · Nature Communications]

Reviewers' Comments:

Reviewer #1:

Remarks to the Author:

Quantum marginal has many applications in Physics and Chemistry. In particular a version of this problem regarding fermions is known to be potentially very important for finding ground states of Hamiltonians. The authors show an interesting relation between the marginal problem and the separability problem. They provide semidefinite programs which can decide whether some given marginals are compatible with some pure global quantum state. Then they show that the existence of multiparticle absolutely maximally entangled states for a given dimension is equivalent to the separability of an explicitly given two-party quantum state.

The paper is well written and I found it interesting to read. Examples are well thought and give additional insight for the reader. Regarding bibliography I think the authors should provide more citation showing recent progress in QMP. For example there were two important paper regarding constructions of states that are on the border of Klyachko polytopes: Christian Schilling et al 2020 New J. Phys. 22 023001 and Tomasz Maciążek et al 2020 New J. Phys. 22 023002 - they in my opinion should be included. Also I think that the approach via semidefinite programs should be compared (at least for the case of 1-representability) with the procedure given by Klyachko in terms of computational complexity. Once this analysis is added I can recommend publication.

Reviewer #2:

Remarks to the Author:

The authors rephrase the quantum marginal problem in terms of a convex optimisation problem for separable quantum states.

To decide whether the maximal value is achieved for a separable state, separability is rephrased in terms of an infinite hierarchy of semi-definite programs.

The connection is applied to the problem to decide the existence of certain quantum error-correcting codes, and so-called AME states as special cases thereof. For AME states, the solution of the optimisation problem can be explicitly formulated, and the remaining problem is to decide whether that state is separable or not.

The approach is clearly interesting and appears to be novel. Its practical values, however, remains unclear. The authors missed to compare the complexity/feasibility to that of other method. Related, it would be interesting to indicate differences in just deciding whether there is a solution or finding one. If one had found a solution to the optimisation problem, it still remains to find a component of the decomposition into a convex combination of pure product states.

In the main text, the authors state "Actually, they can reproduce all the known nonexistence results except $\text{AME}(7, 2)$ [18, 19]."

First, I'd like to suggest to rephrase this as "can reproduce the known nonexistence results from [19] except $\text{AME}(7,2)$ [19]".

More importantly, the authors fail to discuss which of the approaches they consider more simple. In the Supplementary Material, only the case $\text{AME}(7,2)$ is discussed, for which the method proposed in this manuscript was inconclusive.

Overall, I recommend publication of the manuscript, but would like to ask the authors to address the following when preparing a revised version.

- I assume that in the "first key observation" following eq. (5), it should read "There exists a pure

state in \mathcal{C}_2 .

- It is not obvious to me how eq. (42) implies the Singleton bound for pure quantum codes. It would be good to provide more details.
- In eq. (58), I suggest to use brackets for the tensor product ($M_0 \otimes I$)
- I consider the forward reference in the phrase "with the result from Ref. [40]" bad style. One should give at least some hints what type of results are expected to be shown in [40].
- In the Supplementary Material, I suggest to repeat the definition of the parameter r , e.g., in eq. (A4) to make the text more accessible.
- At the end of the Supplementary Material, the authors state "The states ... in Eqs. (E8,E9) are entangled." Supposedly, this conclusion is drawn from the fact that there is no AME(7,2) state.
- As already mentioned, it would be instructive to add an example of an AME state for which the proposed method can be used to show existence/non-existence.

Reviewer #3:

Remarks to the Author:

See comments attached in separate file.

Referee Report for “Complete hierarchy for the quantum marginal problem”

Manuscript Summary

This article demonstrates that “the existence of a pure multipartite state with given marginals can be reformulated as the existence of a two-party separable state with additional semidefinite constraints.” Since the property of separability is itself encodable via a complete semidefinite programming hierarchy, it follows that the pure state quantum marginal problem can therefore be mapped to a complete SDP hierarchy. The authors demonstrate an application of this reformulation by efficiently recovering many known results regarding the nonexistence of certain absolutely maximally entangled (AME) states as well as important known results regarding the nonexistence of certain pure quantum codes. In Theorem 3, the authors reformulate unresolved questions regarding the existence of AME states into (open) problems regarding the separability of specific states. In Theorem 6 the authors map unresolved questions regarding the existence of certain pure quantum codes to problems regarding the existence of *nonspecific* separable states with constrained marginals.

Overall Judgement

The central contribution of this manuscript is an extremely powerful result with potential broad applicability across quantum information theory. There is not a shadow of doubt that this manuscript is worthy of publication. [If anything, I am somewhat surprised to see a result of this magnitude in Nature Communications. A result of this caliber could be submitted directly to Nature, as far as I’m concerned.] Like many of the best ideas in theoretical physics, the main result seems obvious in retrospect. Of course, it is anything *but* obvious. The pure state quantum marginal problem has been around for decades, but only now *finally* has it been recast as an SDP.

This will almost certainly be a very high impact [read: highly cited] paper. The result is of such fundamental importance that I suspect it will soon become part of the standard core curriculum for quantum information theory.

The manuscript could be improved by reorganizing the content to motivate the importance of an SDP formulation more clearly in the introduction, instead of waiting until page 4 to discuss the unique benefits afforded by convexity. The proof of Theorem 1 is somewhat scatted. Some of the elements of the proof appear prior to the Theorem statement, some elements after it, and moreover some critical elements of the proof are relegated to the Methods section. Readers would be better served by a more coherent narrative. I also found the use “Theorem” environment to be overused: the results in this manuscript are notable inconsistent in terms of significance and the scope of applicability.

I will discuss each of these constructive criticisms in more detail below. However, the fundamentals of this manuscript are already quite sound, and there I would grant editorial discretion to consider publishing the manuscript as is without any revision whatsoever.

Comments

1. **[Motivating reformulating as SDP]** To a naïve reader, it may seem odd why the authors are struggling to reformulate the pure state quantum marginal problem as an SDP. Without semidefinite programming, it is quite easy to *mathematically* encode the pertinent problem. Just demand the existence of a ρ satisfying the marginal problem encoded in Eq. (2), and then simply impose $\text{Tr}[\rho^2] = 1$. Everything is analytic, so why bother insisting on a convex optimization reformulation? There are at least two good reasons to want to move to an SDP. Firstly, there is the matter of *computational* implementation. We do not have good software for solving nonconvex optimization (or nonconvex *feasibility*) problems of the naïve sort involving the $\text{Tr}[\rho^2] = 1$ constraint. Practically speaking, such a constraint can not be incorporated within the DCP ruleset. Secondly, there is the tremendous advantage of being able to mod-out symmetry groups when performing convex optimization. That is, it is extremely beneficial to be able to incorporate symmetry constraints such as Eq. (29), as these extra equalities dramatically reduce the number of free variables in the optimization (or feasibility) problem. However, while Eq. (28) holds true for both convex and nonconvex problem formulations, *only* convexity allows the leap from Eq. (28) to Eq. (29). It would be useful to move this discussion up to the introduction, instead of being in the second half of the manuscript.

2. **[Clarify definition of swap operator]** The swap operator is introduced in Eq. (4). I would have found the paper easier to comprehend if there would have been emphasis that the V_{AB} is *not* the conventional swap *superoperator* which exchanges parties A & B. Rather, it is an operator which acts *only* on the Ket space but *not* the Bra space. I would advocate introduce a numbered equation of the sort

$$"V_{AB} := \sum_{ij} |j, i\rangle\langle i, j| \quad \text{such that if } \rho = \sum_{ijkl} \omega_{ijkl} |i, j\rangle\langle k, l|, \text{ then } V_{AB}\rho = \sum_{ijkl} \omega_{ijkl} |j, i\rangle\langle k, l|."$$

3. **[Reorganizing the proof of Theorem 1]** I find the proof of theorem 1 to be oddly scattered. I also find the expression of the Theorem to be counterintuitive. I would advise moving a *variant* of Theorem 1 somewhere in between Eqs. (2) and (3), and then quickly proving that the theorem is true though a few short steps, possibly including the material which is currently relegated to the Methods proof of Lemma 7. Here's a sketch of what I think captures the essence of the argument in a pedagogically optimized version:

$$\begin{aligned} \text{Theorem 1*}: \quad & \text{Tr}[\rho^2] = 1 \text{ and } \forall_{I \in J} \text{Tr}_I[\rho] = \rho_I \text{ if and only if} \\ & \exists \Phi_{AB} \text{ such that } \text{Tr}[V_{AB}\Phi_{AB}] = 1 \text{ where} \\ & \text{Tr}_{A_I c B_I c}[\Phi_{AB}] = \rho_I \otimes \rho_I \text{ and } \Phi_{AB} \in \text{SEP} \text{ and } V_{AB}\Phi_{AB}V_{AB} = \Phi_{AB} \end{aligned}$$

Proof: The existence of such a Φ_{AB} is trivial, as taking $\Phi_{AB} = \rho \otimes \rho$ yields $\text{Tr}[\rho^2] = \text{Tr}[V_{AB}\Phi_{AB}]$. The "if" direction is less trivial. Firstly, we note that the separability constraint implies that $\Phi_{AB} = \sum_u p_u (\sigma^{(u)} \otimes \tau^{(u)})$. On the other hand, $V_{AB}\Phi_{AB}V_{AB} = \Phi_{AB}$ informs us that Φ_{AB} is a two-party state acting on the symmetric subspace only, and hence we must have $\Phi_{AB} = \sum_u p_u (\sigma^{(u)} \otimes \sigma^{(u)})$. Now, from $\text{Tr}_{A_I c B_I c}[\Phi_{AB}] = \rho_I \otimes \rho_I$ we find $\forall_u: \sigma_I^{(u)} = \rho_I$; this follows from the fact that $\forall_u: \text{Tr}[\sigma_I^{(u)}] = 1$ and that $\forall_{X| \text{Tr}[X\rho_I]=0}: \text{Tr}[X\sigma_I^{(u)}] = 0$ per $\text{Tr}[(X \otimes X)(\rho_I \otimes \rho_I)] = 0 = \sum_u p_u \text{Tr}[X\sigma_u]^2$. Accordingly, every *individual* $\sigma^{(u)}$ is a solution to the SDP. Finally, from the well-known relation [30] we have $\text{Tr}[V_{AB} \sum_u p_u (\sigma^{(u)} \otimes \sigma^{(u)})] = \sum_u p_u \text{Tr}[(\sigma^{(u)})^2]$, which equals one only

if $\forall_u: \text{Tr} \left[\left(\sigma^{(u)} \right)^2 \right] = 1$. We therefore have a constructive solution for ρ , namely $\rho = \sigma^{(1)}$, which will satisfy the desired properties of $\text{Tr}[\rho^2] = 1$ and $\forall_{I \in J} \text{Tr}_I \rho = \rho_I$.

Note that I have shifted quite a few things around.

- a. I have incorporated the proof of Lemma 7 directly into the proof of the theorem, instead of relegating it to the methods. Note that also that I have given the proof that

$$\text{Tr}_{A_I c B_I c} \left[\sum_u p_u \left(\sigma^{(u)} \otimes \sigma^{(u)} \right) \right] = \rho_I \otimes \rho_I \quad \vdash \quad \forall_u: \sigma_I^{(u)} = \rho_I \text{ instead of proving that}$$

$$\sum_u p_u \left(\sigma^{(u)} \otimes \sigma^{(u)} \right) = \rho \otimes \rho \quad \vdash \quad \forall_u: \sigma^{(u)} = \rho. \quad \text{I found it personally quite difficult to infer why the latter statement --- i.e. Lemma 7 in the current manuscript --- was relevant to the proof of Theorem 1, until that the authors intended to apply the Lemma to each of the } I \otimes I \text{ subspaces individually.}$$

- b. I have replaced Eq. (9) in the manuscript's version of the theorem with Eq. (12). This makes the proof considerably simpler, in my opinion. The authors can point out to each of the $\text{Tr}_{A_I c B_I c} [\Phi_{AB}] = \rho_I \otimes \rho_I$ can be strengthened into $\text{Tr}_{A_I c} [\Phi_{AB}] = \rho_I \otimes \text{Tr}_A [\Phi_{AB}]$ without loss of generality, as the stronger constraint implies the weaker one under $V_{AB} \Phi_{AB} V_{AB} = \Phi_{AB}$, and the stronger partial trace constraint is clearly also satisfied by $\Phi_{AB} = \rho \otimes \rho$.
- c. I do not invoke the implication of purity until the very last step of the proof. Contrast this with the manuscript's use of Eq. (10) *before* invoking Lemma 7.
- d. I incorporate the condition of symmetry under party relabelling, namely $V_{AB} \Phi_{AB} V_{AB} = \Phi_{AB}$ into the SDP. This has many purposes. More constraints are better; it naturally sets the reader up for the multipartite-symmetry in Theorem 2; it gets us into the symmetric subspace without invoking a derivation from $\text{Tr}[V_{AB} \Phi_{AB}] = 1$ by eigenvalue arguments.
- e. I keep *all* parts of the proof *after* the statement of the Theorem, unlike Eqs. (4) and (5) in the current manuscript.

Long story short, I urge the authors to take another stab at organizing the proof of Theorem 1.

4. **[All theorems are not created equal]** Some theorems are incredibly broad; others feel much more limited. For example, a theorem about encoding the purity of a density matrix as an SDP has wide applicability. The SDP defining the quantum marginal problem (i.e. Theorem 1) is a technically a *derivative* of the purity-to-SDP result (though the current manuscript consciously downplays this). Theorems 6 and 8 pertain to absolutely maximally entangled states and pure quantum codes respectively, which are specific applications of the pure state quantum marginal problem. Calling all these distinct results "Theorem" seems to demean the significance of the earlier results by comparing them to the later results. Moreover, Theorem 6 and Theorem 8 are themselves qualitatively distinct from each other. Theorem 6 is a very constructive problem reformulation: The AME state existence problem is mapped to the problem of determining the separability of *explicit* highly symmetric states. In other words, the Theorem 6 is an *enormously significant simplification* of the AME problem. Theorem 8 seems to just take the *definition* of a pure quantum code and plug it into the pure state quantum marginal problem via Theorem 2. Perhaps I have misunderstood? But Theorem 8 seems nearly an obvious corollary of Theorem 2, whereas Theorem 6 seems far more refined. Calling everything "Theorem" makes these distinctions harder to notice.

5. **[Mention purity in title]** There quantum marginal problem comes in two variants: When the global state is restricted to be pure, and when the global state is unrestricted. The unrestricted case has long since been formulated as an SDP, whereas the contribution of this manuscript is to map the pure state version of the problem to an SDP. Perhaps have the title reflect this? I suggest "A complete hierarchy for the pure state marginal problem." (Note that I also think the article "A" should be prepended.)

Response to Reviewer #1

(Referee's remarks from the referee report are highlighted in blue color.)

Quantum marginal has many applications in Physics and Chemistry. In particular a version of this problem regarding fermions is known to be potentially very important for finding ground states of Hamiltonians. The authors show an interesting relation between the marginal problem and the separability problem. They provide semidefinite programs which can decide whether some given marginals are compatible with some pure global quantum state. Then they show that the existence of multiparticle absolutely maximally entangled states for a given dimension is equivalent to the separability of an explicitly given two-party quantum state.

The paper is well written and I found it interesting to read. Examples are well thought and give additional insight for the reader.

Our Reply: We thank the referee for his/her time and effort invested in reading and evaluating our manuscript. We are glad to see the referee's high judgement of our manuscript.

Regarding bibliography I think the authors should provide more citation showing recent progress in QMP. For example there were two important paper regarding constructions of states that are on the border of Klyachko polytopes: Christian Schilling et al 2020 New J. Phys. 22 023001 and Tomasz Maciążek et al 2020 New J. Phys. 22 023002 - they in my opinion should be included.

Our Reply: We thank the referee for bringing these references to our attention. We have added them in the revised manuscript (Refs. [14, 15]). Please see the first paragraph on page 1.

Also I think that the approach via semidefinite programs should be compared (at least for the case of 1-representability) with the procedure given by Klyachko in terms of computational complexity.

Our Reply: We thank the referee for raising this point. As the quantum marginal problem is known to be QMA-complete (and hence NP-hard), it is to be expected that there exists no algorithm that solves the problem efficiently. Even for the non-overlapping marginals, Klyachko's method quickly becomes computationally intractable in practice due to the large number of marginal inequalities. Indeed, whether there exists a polynomial-time algorithm for this simplest case is still an open problem. Actually, a great benefit of our method is that we can take advantage of the symmetry of the physical system to significantly simplify the cor-

responding problem, as we illustrate with the AME problem in the manuscript. To clarify this point, we have added some discussion on page 4 in the revised manuscript. Please see the paragraph after Theorem 2.

Once this analysis is added I can recommend publication.

Our Reply: We thank the referee for the recommendation. We hope our reply resolves the concerns.

Response to Reviewer #2

(Referee's remarks from the referee report are highlighted in blue color.)

The authors rephrase the quantum marginal problem in terms of a convex optimisation problem for separable quantum states. To decide whether the maximal value is achieved for a separable state, separability is rephrased in terms of an infinite hierarchy of semi-definite programs.

The connection is applied to the problem to decide the existence of certain quantum error-correcting codes, and so-called AME states as special cases thereof. For AME states, the solution of the optimisation problem can be explicitly formulated, and the remaining problem is to decide whether that state is separable or not.

The approach is clearly interesting and appears to be novel.

Our Reply: We thank the referee for his/her time and effort invested in reading and evaluating our manuscript. We are glad to see that the referee appreciates the novelty of our findings.

Its practical values, however, remains unclear. The authors missed to compare the complexity/feasibility to that of other method. Related, it would be interesting to indicate differences in just deciding whether there is a solution or finding one. If one had found a solution to the optimisation problem, it still remains to find a component of the decomposition into a convex combination of pure product states.

Our Reply: We thank the referee for this comment. As the referee rightly points out, there is indeed a difference between the existence of a solution and finding a solution, however, our method still provides significant practical value. On the

one hand, there is no difference if no solution exists, i.e., when the method excludes the possibility of a solution. For instance, we recover almost all known AME nonexistence results [IEEE Trans. Inf. Theory 44, 1369 (1998), Phys. Lett. A 273, 213 (2000), Phys. Rev. A 69, 052330 (2004), J. Phys. A: Math. Theor. 51, 175301 (2018)] with a single unified approach. Furthermore, in contrast to existing methods, there is no fundamental limit to our approach; it, in principle, can be used to resolve the AME existence problem for any number of parties and local dimensions. On the other hand, the proof of Theorem 1 implies that any decomposition of the separable state provides an explicit solution to the marginal problem. For the decomposition of separable states, many known algorithms exist, such as Refs. [Phys. Rev. Lett. 120, 050506 (2017), Phys. Rev. Lett. 103, 160404 (2009), Phys. Rev. A 76, 032318 (2007)], which can help finding a pure state with the desired marginals. To clarify this point, we added a discussion on page 3 in the revised manuscript. Please see the paragraph after Eqs. (14,15,16). Also, in the paragraph after Theorem 2 on page 4, we added a short discussion on the general complexity of our approach, stressing that it allows for a straight-forward implementation of symmetries reducing the complexity significantly for certain problems.

In the main text, the authors state "Actually, they can reproduce all the known nonexistence results except AME(7, 2) [18, 19]." First, I'd like to suggest to rephrase this as "can reproduce the known nonexistence results from [19] except AME(7,2) [19]". More importantly, the authors fail to discuss which of the approaches they consider more simple. In the Supplementary Material, only the case AME(7,2) is discussed, for which the method proposed in this manuscript was inconclusive.

Our Reply: We thank the referee for the suggestion. We have rephrased the sentence as "Actually, they can reproduce all the known nonexistence results [30] except AME(7,2) [20]." As far as we know, Ref. [30] (an up-to-date online table published in accompany with Ref. [21; previously 19]) provides a complete collection of all results on the AME existence problem which is why we leave out the word "from" before this reference. Furthermore, we added the explicit example of AME(4, 2) to the Supplementary Material. Please see the paragraph after Eq. (39) on page 5 of the main text and the discussion around Eq. (25) in Supplementary Note 2. Finally, see the reply to the previous comment above about the comparison to known methods.

Overall, I recommend publication of the manuscript, but would like to ask the authors to address the following when preparing a revised version.

Our Reply: We thank the referee for recommending the publication of our manuscript. The concerns are addressed in the following.

- I assume that in the "first key observation" following eq. (5), it should read "There exists a pure state in \mathcal{C}_2 ".

Our Reply: Indeed, we willingly phrase this observation the way we did to stress that the original pure state marginal problem can be solved considering the two-copy system. This is the essential connection that the key observation provides. Hence, we would like to keep the current statement because the set \mathcal{C} is the set of global states that we are interested in for the quantum marginal problem.

- It is not obvious to me how eq. (42) implies the Singleton bound for pure quantum codes. It would be good to provide more details.

Our Reply: We thank the referee for this comment. As $|Q\rangle$ is a pure state in $\tilde{\mathcal{H}} = \mathcal{H}_{0I} \otimes \mathcal{H}_{I^c}$ where $\mathcal{H}_{0I} = \mathcal{H}_0 \otimes \mathcal{H}_I$, the reduced state $\rho_{0I} := \text{Tr}_{I^c}(|Q\rangle\langle Q|)$ satisfies $\text{rank}(\rho_{0I}) \leq \min\{\dim(\mathcal{H}_{0I}), \dim(\mathcal{H}_{I^c})\} \leq \sqrt{\dim(\tilde{\mathcal{H}})}$, i.e., $Kd^m \leq \sqrt{Kd^n}$, from which the Singleton bound for pure quantum codes follows. To clarify this point, we have added a few sentences on page 6 in the revised manuscript. Please see the paragraph before the Discussion section.

- In eq. (58), I suggest to use brackets for the tensor product ($M_0 \otimes I$)

Our Reply: We thank the referee for this suggestion. We have added the brackets in the revised manuscript.

- I consider the forward reference in the phrase "with the result from Ref. [40]" bad style. One should give at least some hints what type of results are expected to be shown in [40].

Our Reply: Following the suggestion of the referee, we changed the phrase "with the result from Ref. [40]" to "with the general result on rank-constrained optimization from Ref. [48]" and the title of the paper in preparation has also been added to the References in the revised manuscript.

- In the Supplementary Material, I suggest to repeat the definition of the parameter r , e.g., in eq. (A4) to make the text more accessible.

Our Reply: We thank the referee for the suggestion. The definition of the parameter r has been added to the revised Supplementary Material.

- At the end of the Supplementary Material, the authors state "The states ... in Eqs. (E8,E9) are entangled." Supposedly, this conclusion is drawn from the fact

that there is no AME(7,2) state.

Our Reply: Indeed, it is true that this conclusion is drawn from the nonexistence of AME(7, 2) state. We have added the sentence “As no AME(7, 2) state exists” to clarify this point in the revised Supplementary Material.

- As already mentioned, it would be instructive to add an example of an AME state for which the proposed method can be used to show existence/nonexistence.

Our Reply: We thank the referee for this suggestion. The example of AME(4, 2) has been added to the revised Supplementary Material.

Response to Reviewer #3

(Referee’s remarks from the referee report are highlighted in blue color.)

Manuscript Summary

This article demonstrates that “the existence of a pure multiparticle state with given marginals can be reformulated as the existence of a two-party separable state with additional semidefinite constraints.” Since the property of separability is itself encodable via a complete semidefinite programming hierarchy, it follows that the pure state quantum marginal problem can therefore be mapped to a complete SDP hierarchy. The authors demonstrate an application of this reformulation by efficiently recovering many known results regarding the nonexistence of certain absolutely maximally entangled (AME) states as well as important known results regarding the nonexistence of certain pure quantum codes. In Theorem 3, the authors reformulate unresolved questions regarding the existence of AME states into (open) problems regarding the separability of specific states. In Theorem 6 the authors map unresolved questions regarding the existence of certain pure quantum codes to problems regarding the existence of nonspecific separable states with constrained marginals.

Overall Judgement

The central contribution of this manuscript is an extremely powerful result with potential broad applicability across quantum information theory. There is not a shadow of doubt that this manuscript is worthy of publication. [If anything, I am somewhat surprised to see a result of this magnitude in Nature Communications.

A result of this caliber could be submitted directly to Nature, as far as I'm concerned.] Like many of the best ideas in theoretical physics, the main result seems obvious in retrospect. Of course, it is anything but obvious. The pure state quantum marginal problem has been around for decades, but only now finally has it been recast as an SDP.

This will almost certainly be a very high impact [read: highly cited] paper. The result is of such fundamental importance that I suspect it will soon become part of the standard core curriculum for quantum information theory.

The manuscript could be improved by reorganizing the content to motivate the importance of an SDP formulation more clearly in the introduction, instead of waiting until page 4 to discuss the unique benefits afforded by convexity. The proof of Theorem 1 is somewhat scatted. Some of the elements of the proof appear prior to the Theorem statement, some elements after it, and moreover some critical elements of the proof are relegated to the Methods section. Readers would be better served by a more coherent narrative. I also found the use "Theorem" environment to be overused: the results in this manuscript are notable inconsistent in terms of significance and the scope of applicability.

I will discuss each of these constructive criticisms in more detail below. However, the fundamentals of this manuscript are already quite sound, and there I would grant editorial discretion to consider publishing the manuscript as is without any revision whatsoever.

Our Reply: We thank the referee for his/her time and effort invested in reading and evaluating our manuscript. We are glad to see the referee's high judgement of our manuscript and his/her support for the publication of our manuscript in Nature Communications. We also thank the referee for the constructive comments on our manuscript, which helped a lot to improve our manuscript.

Comments

1. **[Motivating reformulating as SDP]** To a naïve reader, it may seem odd why the authors are struggling to reformulate the pure state quantum marginal problem as an SDP. Without semidefinite programming, it is quite easy to mathematically encode the pertinent problem. Just demand the existence of a ρ satisfying the marginal problem encoded in Eq. (2), and then simply impose $\text{Tr}(\rho^2) = 1$. Everything is analytic, so why bother insisting on a convex optimization reformulation? There are at least two good reasons to want to move to an SDP. Firstly, there is the matter of computational implementation. We do not have good software for solving nonconvex optimization (or nonconvex feasibility) problems of the naïve sort involving the $\text{Tr}(\rho^2) = 1$ constraint. Practically speaking, such a constraint can not be incorporated within the DCP ruleset. Secondly, there is the tremendous

advantage of being able to mod-out symmetry groups when performing convex optimization. That is, it is extremely beneficial to be able to incorporate symmetry constraints such as Eq. (29), as these extra equalities dramatically reduce the number of free variables in the optimization (or feasibility) problem. However, while Eq. (28) holds true for both convex and nonconvex problem formulations, only convexity allows the leap from Eq. (28) to Eq. (29). It would be useful to move this discussion up to the introduction, instead of being in the second half of the manuscript.

Our Reply: We thank the referee for this suggestion. It is indeed very helpful to mention the motivations for reformulating the quantum marginal problem as SDPs. Following the referee’s suggestion, we have added a few sentences in the introduction to clarify the motivations. Please see the last paragraph on page 1 and the first two paragraphs on page 2 in the revised manuscript.

2. [Clarify definition of swap operator] The swap operator is introduced in Eq. (4). I would have found the paper easier to comprehend if there would have been emphasis that the VAB is not the conventional swap superoperator which exchanges parties A & B. Rather, it is an operator which acts only on the Ket space but not the Bra space. I would advocate introduce a numbered equation of the sort “ $V_{AB} = \sum_{ij} |j, i\rangle \langle i, j|$ such that if $\rho = \sum_{ijkl} \omega_{ijkl} |i, j\rangle \langle k, l|$, then $V_{AB}\rho = \sum_{ijkl} \omega_{ijkl} |j, i\rangle \langle k, l|$.”

Our Reply: We thank the referee for this comment. We have added a numbered equation to explicitly show the definition of the swap operator V_{AB} . Please see Eq. (5) in the revised manuscript.

3. [Reorganizing the proof of Theorem 1] I find the proof of theorem 1 to be oddly scattered. I also find the expression of the Theorem to be counterintuitive. I would advise moving a variant of Theorem 1 somewhere in between Eqs. (2) and (3), and then quickly proving that the theorem is true though a few short steps, possibly including the material which is currently relegated to the Methods proof of Lemma 7. Here’s a sketch of what I think captures the essence of the argument in a pedagogically optimized version:

Theorem 1*: $\text{Tr}(\rho^2) = 1$ and $\forall I \in \mathcal{I} \text{Tr}_I[\rho] = \rho_I$ if and only if $\exists \Phi_{AB}$ such that $\text{Tr}[V_{AB}\Phi_{AB}] = 1$ where $\text{Tr}_{A^c B^c}[\Phi_{AB}] = \rho_I \otimes \rho_I$ and $\Phi_{AB} \in \text{SEP}$ and $V_{AB}\Phi_{AB}V_{AB} = \Phi_{AB}$.

Proof: The existence of such a Φ_{AB} is trivial, as taking $\Phi_{AB} = \rho \otimes \rho$ yields $\text{Tr}[\rho^2] = \text{Tr}[V_{AB}\Phi_{AB}]$. The “if” direction is less trivial. Firstly, we note that the separability constraint implies that $\Phi_{AB} = \sum_u p_u \left(\sigma^{(u)} \otimes \tau^{(u)} \right)$. On the other hand, $V_{AB}\Phi_{AB}V_{AB} = \Phi_{AB}$ informs us that Φ_{AB} is a two-party state acting on the sym-

metric subspace only, and hence we must have $\Phi_{AB} = \sum_u p_u \left(\sigma^{(u)} \otimes \sigma^{(u)} \right)$. Now, from $\text{Tr}_{A_I^c B_I^c} [\Phi_{AB}] = \rho_I \otimes \rho_I$ we find $\forall_u : \sigma_I^{(u)} = \rho_I$; this follows from the fact that $\forall_u : \text{Tr} \left[\sigma_I^{(u)} \right] = 1$ and that $\forall_{X | \text{Tr}(X\rho_I)=0} : \text{Tr} \left[X\sigma_I^{(u)} \right] = 0$ per $\text{Tr}[(X \otimes X)(\rho \otimes \rho)] = 0 = \sum_u p_u \text{Tr} [X\sigma_u]^2$. Accordingly, every *individual* $\sigma^{(u)}$ is a solution to the SDP. Finally, from the well-known relation [30] we have $\text{Tr} \left[V_{AB} \sum_u p_u \left(\sigma^{(u)} \otimes \sigma^{(u)} \right) \right] = \sum_u p_u \text{Tr} \left[\left(\sigma^{(u)} \right)^2 \right]$, which equals one only if $\forall_u : \text{Tr} \left[\left(\sigma^{(u)} \right)^2 \right] = 1$. We therefore have a constructive solution for ρ , namely $\rho = \sigma^{(1)}$, which will satisfy the desired properties of $\text{Tr}(\rho^2) = 1$ and $\forall_{I \in \mathcal{I}} \text{Tr}_{I^c}[\rho] = \rho_I$.

Our Reply: We thank the referee for the detailed suggestion for reorganizing the proof of Theorem 1. We have followed most of the suggestions of the referee and reorganized the proof of Theorem 1. The detailed changes and some minor differences are shown below.

Note that I have shifted quite a few things around.

a. I have incorporated the proof of Lemma 7 directly into the proof of the theorem, instead of relegating it to the methods. Note that also that I have given the proof that $\text{Tr}_{A_I^c B_I^c} \left[\sum_u p_u \left(\sigma^{(u)} \otimes \sigma^{(u)} \right) \right] = \rho_I \otimes \rho_I \quad \vdash \quad \forall_u : \sigma_I^{(u)} = \rho_I$ instead of proving that $\sum_u p_u \left(\sigma^{(u)} \otimes \sigma^{(u)} \right) = \rho \otimes \rho \quad \vdash \quad \forall_u : \sigma^{(u)} = \rho$. I found it personally quite difficult to infer why the latter statement — i.e. Lemma 7 in the current manuscript — was relevant to the proof of Theorem 1, until that the authors intended to apply the Lemma to each of the $I \otimes I$ subspaces individually.

Our Reply: We followed the referee’s suggestion and incorporated the proof of Lemma 7 directly into the proof. Accordingly, Lemma 7 is removed from the Method section.

b. I have replaced Eq. (9) in the manuscript’s version of the theorem with Eq. (12). This makes the proof considerably simpler, in my opinion. The authors can point out to each of the $\text{Tr}_{A_I^c B_I^c} [\Phi_{AB}] = \rho_I \otimes \rho_I$ can be strengthened into $\text{Tr}_{A_I^c} [\Phi_{AB}] = \rho_I \otimes \text{Tr}_A [\Phi_{AB}]$ without loss of generality, as the stronger constraint implies the weaker one under $V_{AB} \Phi_{AB} V_{AB} = \Phi_{AB}$, and the stronger partial trace constraint is clearly also satisfied by $\Phi_{AB} = \rho \otimes \rho$.

Our Reply: Following the suggestion, we rephrased Theorem 1 and the proof with the constraint $\text{Tr}_{A_I^c B_I^c} [\Phi_{AB}] = \rho_I \otimes \rho_I$. The stronger constraint $\text{Tr}_{A_I^c} [\Phi_{AB}] = \rho_I \otimes \text{Tr}_A [\Phi_{AB}]$ is moved to the remarks after the proof.

c. I do not invoke the implication of purity until the very last step of the proof. Contrast this with the manuscript's use of Eq. (10) before invoking Lemma 7.

Our Reply: Concerning this point, we would like to keep the current form of Eq. (10), as it comes from a well-known result from entanglement theory and can be directly generalized to the multipartite case [Phys. Rev. Lett. 102, 170503 (2009)]. Another reason is shown in the following reply.

d. I incorporate the condition of symmetry under party relabelling, namely $V_{AB}\Phi_{AB}V_{AB} = \Phi_{AB}$ into the SDP. This has many purposes. More constraints are better; it naturally sets the reader up for the multipartite-symmetry in Theorem 2; it gets us into the symmetric subspace without invoking a derivation from $\text{Tr}(V_{AB}\Phi_{AB}) = 1$ by eigenvalue arguments.

Our Reply: We thank the referee for this suggestion, however, the symmetry condition $V_{AB}\Phi_{AB}V_{AB} = \Phi_{AB}$ is not sufficient to guarantee that Φ_{AB} is within the symmetric subspace. In particular, the constraint $V_{AB}\Phi_{AB}V_{AB} = \Phi_{AB}$ does not imply that a separable Φ_{AB} is of the form $\Phi_{AB} = \sum_u p_u (\sigma^{(u)} \otimes \sigma^{(u)})$. An explicit counterexample is given by the state $\Phi_{AB} = |0\rangle\langle 0|_A \otimes |1\rangle\langle 1|_B + |1\rangle\langle 1|_A \otimes |0\rangle\langle 0|_B$, because $\text{Tr}(V_{AB}\Phi_{AB}) = 0$ which is impossible for Φ_{AB} of the aforementioned form. Hence, we would like to keep the argument that the condition $\text{Tr}(V_{AB}\Phi_{AB}) = 1$ implies the purity in Eq. (10), and later strengthen it to the condition $V_{AB}\Phi_{AB} = \Phi_{AB}$.

e. I keep all parts of the proof after the statement of the Theorem, unlike Eqs. (4) and (5) in the current manuscript.

Long story short, I urge the authors to take another stab at organizing the proof of Theorem 1.

Our Reply: We thank the referee again for the constructive suggestion. Now, the proof of Theorem 1 is reorganized to be self-contained. The Eqs. (4, 5) [Eqs. (4, 6) in the revised manuscript] are kept as the motivation for introducing the swap operator.

4. [All theorems are not created equal] Some theorems are incredibly broad; others feel much more limited. For example, a theorem about encoding the purity of a density matrix as an SDP has wide applicability. The SDP defining the quantum marginal problem (i.e. Theorem 1) is a technically a derivative of the purity-to-SDP result (though the current manuscript consciously downplays this). Theorems 6 and 8 pertain to absolutely maximally entangled states and pure quantum codes respectively, which are specific applications of the pure state quantum marginal

problem. Calling all these distinct results “Theorem” seems to demean the significance of the earlier results by comparing them to the later results. Moreover, Theorem 6 and Theorem 8 are themselves qualitatively distinct from each other. Theorem 6 is a very constructive problem reformulation: The AME state existence problem is mapped to the problem of determining the separability of explicit highly symmetric states. In other words, the Theorem 6 is an enormously significant simplification of the AME problem. Theorem 8 seems to just take the definition of a pure quantum code and plug it into the pure state quantum marginal problem via Theorem 2. Perhaps I have misunderstood? But Theorem 8 seems nearly an obvious corollary of Theorem 2, whereas Theorem 6 seems far more refined. Calling everything “Theorem” makes these distinctions harder to notice.

Our Reply: We thank the referee for this comment. From the description, we assume the referee was referring to Theorems 3 and 6 (instead of Theorems 6 and 8). We agree with the referee on that calling everything “Theorem” may demean the significance of the earlier results by comparing them to the later results. In the revised manuscript, we still use the term “Theorem” for Theorems 1, 2, and 3, because Theorems 1 and 2 are the main results of the current manuscript and Theorem 3 makes a significant contribution to the field of AME states. Theorem 6 is changed to a Proposition, as it directly follows from our observation that the existence of pure quantum codes can be written as quantum marginal problems (Lemma 5) together with Theorem 1. Similarly, Theorem 9 is also changed to a Proposition.

5. [Mention purity in title] There quantum marginal problem comes in two variants: When the global state is restricted to be pure, and when the global state is unrestricted. The unrestricted case has long since been formulated as an SDP, whereas the contribution of this manuscript is to map the pure state version of the problem to an SDP. Perhaps have the title reflect this? I suggest “A complete hierarchy for the pure state marginal problem.” (Note that I also think the article “A” should be prepended.)

Our Reply: We thank the referee for his/her suggestion. We have changed the title to “A complete hierarchy for the pure state marginal problem in quantum mechanics”. The phrase “in quantum mechanics” is added to emphasize that our result is about the quantum marginal problem instead of the classical marginal problem. This also makes the subject of the manuscript clearer for a multidisciplinary journal such as Nature Communications.

Reviewers' Comments:

Reviewer #1:

Remarks to the Author:

I am satisfied with the modifications. I recommend publication.

Reviewer #2:

Remarks to the Author:

I appreciate the revision of the manuscript by the authors and the clarifications.

I still support publication of the manuscript, but I would like the authors to consider the following when preparing the final version.

The authors introduce Eq. (13) as a "stronger" condition. At the same time, it is argued that the original condition in Eq. (9) was implied when additionally considering Eq. (10).

Hence, just comparing Eq. (9) and Eq. (13), it is not clear to me in which sense Eq. (13) was stronger.

Also, at the end of the paragraph the authors write "the latter constraint may be stronger than the former." There is some ambiguity what "latter and former constraint" actually refer to. Are those Eq. (13) and Eq. (9), respectively. Further, the authors write "may be stronger", while Eq. (13) had been introduced as being stronger.

Response to Reviewer #2

The referee writes:

I appreciate the revision of the manuscript by the authors and the clarifications. I still support publication of the manuscript, but I would like the authors to consider the following when preparing the final version.

The authors introduce Eq. (13) as a "stronger" condition. At the same time, it is argued that the original condition in Eq. (9) was implied when additionally considering Eq. (10). Hence, just comparing Eq. (9) and Eq. (13), it is not clear to me in which sense Eq. (13) was stronger. Also, at the end of the paragraph the authors write "the latter constraint may be stronger than the former." There is some ambiguity what "latter and former constraint" actually refer to. Are those Eq. (13) and Eq. (9), respectively. Further, the authors write "may be stronger", while Eq. (13) had been introduced as being stronger.

Our reply:

We thank the referee for supporting publication of our manuscript and also for the comment which really helped to clarify why Eq. (13) is a stronger condition. Indeed, we do not need Eq. (10) to prove that Eq. (13) is stronger than Eq. (9). Also, we rephrased "the latter constraint may be stronger" to "Eq. (13) may be strictly stronger for certain marginal problems" to avoid the ambiguity. Please see the second paragraph on page 3.